# Revealing Unintentional Information Leakage in Low-Dimensional Facial Portrait Representations

## Abstract

We evaluate the information that can unintentionally leak into the low dimensional output of a neural network, by reconstructing an input image from a 40- or 32-element feature vector that intends to only describe abstract attributes of a facial portrait, in contrast to face embeddings which are specifically designed to encode distinct features. The reconstruction uses blackbox-access to the image encoder which generates the feature vector. Other than previous work, we leverage recent knowledge about image generation and facial similarity, implementing a method that outperforms the current state-of-the-art. Our strategy uses a pretrained Style-GAN and a new loss function that compares the perceptual similarity of portraits by mapping them into the latent space of a FaceNet embedding. Additionally, we present a new technique that fuses the output of an ensemble, to deliberately generate specific aspects of the recreated image.

Despite the increasing ubiquity of AI solutions, the relevance of privacy in the context of machine learning is often overlooked. One important question is of how much unwanted information can leak into an attribute feature vector. Neural networks are applied to extract, e.g., the stress level of drivers (Gao et al., 2014), the engagement of students (Bosch et al., 2016; Whitehill et al., 2014), a medical diagnosis (Esteva et al., 2017) and abstract avatars[1] from photos of users or patients. They are even proposed for explicitly anonymizing private data (Osia et al., 2018; Wu et al., 2018; 2020; Ren et al., 2018; Roy & Boddeti, 2019; Ding et al., 2020; Wang et al., 2018; Raval et al., 2017; Bertran et al., 2019; Pittaluga et al., 2019b). By most of these examples, it is neglected that the information processing of neural networks is not well enough understood to be able to guarantee the absence of private input information left in their output. The information that is supposed to be removed could still be a part of the feature vector, only in an altered way. By making use of additional information it might still be accessible.

This paper aims to raise awareness for the fact that we are currently unable to prevent private input information from leaking into the output of a neural network. We reconstruct an input image from a feature vector which should only describe a few attributes of the image. We train a *decoder* (depicted in Figure 1) which reverts from a feature vector back to the input that created it, focusing on recreating components of the input that should not be a part of the feature vector. The focus of this paper is set on reconstructing human portraits, with the main target on their identity. To achieve such a reconstruction, several challenges need to be overcome by the decoder. One is the nature of the mapping from image to feature vector: the relation is not bijective, more than one input can lead to the same output. To counter this problem, we leverage a pretrained StyleGAN (Karras et al., 2020) image generator to introduce a strong bias towards a defined target distribution. Another difficulty arises from the fact that the exact target of our reconstruction can be hard to define. An optimal decoder would recreate every pixel of the input, which is a complex task, even given our limited search space. However, this is not necessary. A reconstruction that only incorporates the important parts of the input can be just as harmful - for example, a photo of a person with the same identity, taken from a different angle with a different facial expression or background. Hence, optimizing only the pixel-wise error is not a clever approach, and might even prefer a reconstruction of the wrong person in front of the right background. Instead, we map both input and reconstructed images into different latent spaces and compare their embeddings.

---

[1]One example is the *Bitmoji* created by the *Snapchat* app, see `https://www.bitmoji.com/`

The combination of various specialized loss functions improves the reconstruction, but also introduces a third challenge: how to blend the result of decoders trained for different goals? We found that combining the losses into one enormous cost function oftentimes keeps the model from converging - even though all are theoretically aiming for the same perfect solution (an exact reconstruction). A reason might be that the paths of the cost functions during learning are too dissimilar. A solution presented in this paper is to make use of a special property of the StyleGAN image generator: each layer of the image generator creates a certain aspect of the image, like the shape of the face or the pose of the depicted person. By feeding specialized inputs to the individual StyleGAN layers, we can systematically define each component of the generated image.

## 1 RELATED WORK

There are many angles from which machine learning models can be attacked. The topic of this paper falls into the category of data reconstruction at inference time. This line of research can be split according to three properties: (i) **knowledge of the attacker**, who is either given access to all model parameters (whitebox) or only allowed to pose queries to the model and receive an answer, with no information about how that answer was calculated (blackbox), (ii) the **the goal of the attack**: either to infer knowledge about the data used to train a model or to reconstruct a specific input, given an output, with the input not having been part of the training set, and (iii) the **type of encoder**: an encoder trained to extract features from a given image of a person without aiming at being able to reveal the person's identity or a facial embedding network with features explicitly created to encode the identity of a depicted person. A table sorting the most relevant related work into those categories can be found in the Appendix (Table 3).

In the following, we pay particular attention to the two different attack goals. The two goals are similar enough to motivate coinciding strategies but are confronted with different limitations. To the best of our knowledge, the recreation of training data as an attack goal has only been done for encoders that classify whole identities - one class corresponding to exactly one individual. By design, as many features as possible are covered that identify a facial image. The challenge has been tackled from a whitebox (Simonyan et al., 2014; Fredrikson et al., 2015; Zhang et al., 2019; Chen et al., 2020) and blackbox (Kahla et al., 2022; Dionysiou et al., 2023; Han et al., 2023) access, sometimes making a similar use of image generators as our method. A detailed assessment of the most interesting publications is given in Section C.

The most popular targets for the second goal, a feature vector reconstruction attack, are face embeddings: high-level representations of a face or identity. Again, by design, face embedding networks are trained to condense as much information as possible into the embedding vector for person identification.Deng et al. (2019), for example, even use the reconstruction capability as a quality measure for the embedding. Many blackbox attacks (Vendrow & Vendrow, 2021; Dong et al., 2023; Mai et al., 2019; Shahreza et al., 2022; Mai et al., 2019; Dong et al., 2021; Duong et al., 2020) expand upon this goal, proving that if an encoder network is trained to create a condensed version of an identity, this version can be decompressed to reveal most of the original input. By now, major companies[2] have started to recognize this fact and carry out measures to protect such embeddings.

Our research faces a more difficult challenge by reconstructing from feature vectors that do not explicitly aim to keep the person's identity. A typical face embedding is made up of 512 or more elements (Schroff et al., 2015), the attribute vectors that we are reconstructing from contain 40 and 32 items, describing abstract attributes. A large part of the information from the input image is not required to solve the given task, and should therefore, in theory, have been discarded in early layers of the network - giving reason for the intuitive assumption that they are "safe". We are not the first to implement a reconstruction from such attribute vectors. In 2015, 2016, and 2018, respectively, Mahendran & Vedaldi (2015), Dosovitskiy & Brox (2016b) and Nash et al. (2018) published results on the reconstruction of images from different feature representations, like HOG and SIFT descriptors, and the feature maps of a (very small) CNN. All assume full access to the training dataset, but put their focus on understanding network properties, rather than attacking it. Mahendran & Vedaldi (2015) iteratively optimize an input image, using gradient descent through

---

[2]For example Microsoft (`https://learn.microsoft.com/en-us/windows-hardware/design/device-experiences/windows-hello-enhanced-sign-in-security` in September 2023) and Apple (`https://support.apple.com/en-us/102381` in September 2023).

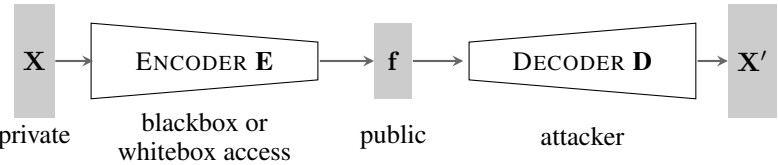

Figure 1: The scenario of this paper: an encoder $E$ creates an attribute vector $f$ for an image, which is reconstructed by our decoder $D$.

the target to minimize a combination of encoder loss and a custom *image prior* term. An image prior can be as simple as the norm of the mean-subtracted image, encouraging the image to stay within a target interval. Their method represents a very early stage of what evolved into the utilization of image generators: implement additional knowledge about the overall nature of our target images into the loss.

Dosovitskiy & Brox (2016b) train a decoder with an inverted dataset: for a set of input images of the same type as the target image set (e.g. facial images), the feature vectors that $E$ returns are determined, and these returned feature vectors then serve as input to $D$. The images define the optimal output of $D$. The authors follow up with an improved loss function shortly after (Dosovitskiy & Brox, 2016a), based on the image- and feature map difference, supplemented by the output of a (very simplistic) discriminator, that is trained alongside the decoder to create more realistic images. The (at the time) innovative use of discriminators makes the images much less blurry.

The same inverted dataset strategy (with no discriminator loss term) achieves better results with an improved decoder architecture in 2019 (Yang et al., 2019). In a similar method Zhao et al. (2021) make use of additional model explanations as a second input to their decoder to improve the accuracy of their reconstruction.

## 2 THREAT MODEL

As depicted in Figure 1, our attack aims at an encoder $E$ that takes as input an image (in our case a portrait of a person) and returns an output $f$ describing certain properties of the input image (e.g. an assessment of the emotion of a patient). The vector $f$ is considered safe, and shared (for example with the intent of scientific evaluation), without linking it to a specific person.

An attacker gains access to $f$, has blackbox knowledge about $E$, and uses additional assumptions about the kind of input image (facial portraits) to find the identity of the depicted person (the patient). It would allow them to leak exceedingly private personal information about their victim.

Generally, our scenario involves three datasets: one used to train $E$, one used by the attacker to train $D$, and data used at inference time, that was neither seen by $E$ nor $D$. We consider two possibilities for the access to those datasets: **(A)**, the target encoder is trained on images from a public dataset. Then this public dataset can also be used to train $D$, and the challenge is to reconstruct images that were never seen by either $D$ or $E$; and **(B)**, the target is trained on a private dataset. The attacker has no access to this dataset and has to use a more or less good substitute, which makes everything much more challenging.

## 3 PROPOSED METHOD

Our method is built around the idea of training a small mapping network to translate a feature vector into a generator input, whereas the strength of the mapping network does not come from its architecture, but rather the sophisticated training strategy.

### 3.1 LEVERAGING IMAGE GENERATORS TO LIMIT THE SEARCH SPACE

The task of reconstructing an image from a low-dimensional feature vector is fundamentally ill-posed. Goodfellow et al. (2015) have shown that adding an imperceptibly small vector to an image

can change the created feature vector significantly - two images that are visually the same create different outputs. Vice versa, a single feature vector could have been created by different images.

While we are unable to enforce a bijective relation between image and feature vector, the challenge can be significantly simplified by making sure that our reconstructed images are from a similar distribution as the images that the target encoder has been trained on. If we can define a set of constraints that all reconstructed images should adhere to (e.g. "facial portraits"), we can greatly limit the search space. To infuse that knowledge into the reconstruction pipeline, our approach is based on two steps: (1) train an image generator to create images that satisfy the known limitations and (2) freeze the generator and train a second network to search over the input space of our trained generator. Since a well-trained generator introduces a very strong bias towards a specific type of image, we are adding the same kind of bias to our reconstruction task.

## 3.2 PGE-TRAINING

Our decoder is now made up of two components: $G$, a pretrained StyleGAN, and $P$, a pre-layer that maps a feature vector to a generator input. Figure 2 illustrates the relation between the three networks involved in the reconstruction. Note that we are only actively training $P$, both $G$ and $E$ remain frozen, and $E$ can only be accessed as a blackbox.

The canonical way of training $P$ would be to generate random noise vectors $n$ drawn from a Gaussian distribution, feed $n$ into G, then E, and determine the $f$ that corresponds to $n$ and minimize the squared distance between both. A significant advantage would be that we can generate an infinite number of samples, and iterate very quickly through large amounts of training data since the training only involves calculating the gradient for the very small network $P$. However, the distribution modeled by G will likely deviate from the specific distribution of $X$. In addition, even the very advanced StyleGAN2 sometimes generates images that are not realistic faces. Therefore, we use what we call *image based training*, which ensures that real face images are used and allows us to control the training data distribution:

1. Select a dataset that is as close as possible to the target dataset[3],

2. For each image $X$ in the dataset, retrieve the matching feature vector $f$ returned by $E$, to create a new dataset $f \rightarrow X$,

3. Train P on this dataset. To evaluate its output $n$, it needs to be mapped into the image space, using $G$, with the loss defined as $L(X, G(P(f)))$ for a distance measure $L$.

The strategy comes with the obvious disadvantage that even though $G$ remains frozen, the gradient needs to be propagated through $G$, making the training slower. We are also once again limited to the number of samples of our training dataset. Additionally, $P$ is no longer encouraged to create $n$ according to Gaussian noise. This difficulty can be largely remedied by adding a term to the overall loss function which we call *distribution loss*

$$L_{\text{dist}}(n) = \frac{1}{|n|} \sum_{n_i \in n} n_i + |1 - \sigma(n)|, \tag{1}$$

which at least encourages distributions of $n$ with zero mean and standard deviation $\sigma = 1$.

The drawbacks of image-based training are nevertheless outweighed by its advantages: the training is bound to a specific dataset - if $G$ generates unrealistic images, they are punished by the loss function that compares the output of $G$ to images from our training set, discouraging any $n$ that would make $G$ create such an image while pushing for $n$ that can reflect the full diversity of the training set. We are further directly comparing images instead of noise, allowing for the advanced loss function presented in Section 3.3.

Note, that we are not using the StyleGAN generator trained with the original method, but a version that is architecturally the same, but trained slightly differently as proposed in Zhao et al. (2020). The training leads to generators which are more robust to input that does not perfectly adhere to a Gaussian distribution.

---

[3]In our experiments, this is always the same dataset that G was trained on

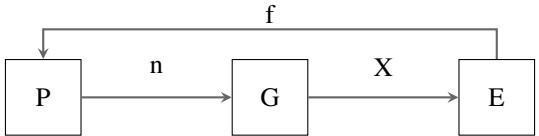

Figure 2: The in- and outputs of the three components of the reconstruction task: the encoder $E$ translates an image $X$ into a feature vector $f$, the pre-generator $P$ creates a generator input from the feature vector, and the generator $G$ turns the input into an image. The combination of $P$ and $G$ makes up the decoder. As described in Section 3.2, our training method is based on keeping $G$ frozen, but nonetheless comparing not $n$, but $X$ to a defined target.

### 3.3 IMAGE SIMILARITY LOSS

The basic loss function for the image-based training (Section 3.2) simply compares the pixels of the original and the recreated image. For $n = P(f)$, obtained for the feature vector $f = E(X)$, the pixel-wise loss $L_{\mathrm{pixel}}$ is given by

$$L_{\mathrm{pixel}}(n) = (X - G(n))^2. \tag{2}$$

where $G$ is the image of the pretrained image generator (see Figure 2).

However, the goal of our attack is not to recreate every pixel but to identify the person seen in the image. Our training needs to focus on the information of the image that is relevant for a human to identify the depicted person, which is not easily defined. A reconstruction of a face that is taken from a different angle could still be identifiable. Contrarily, if a reconstruction gets the exact pixel values of 95% of the target image perfectly right, but fails for a few small, but significant areas, like the shape of the eyes, the depicted person could be unrecognizable. To train explicitly for our defined goal, we are instead comparing the embedding created by the FaceNet (Schroff et al., 2015) network. Both target image and recreation are given to the embedding network, and the loss is given by the squared distance between them:

$$L_{\mathrm{facenet}}(n) = (F(X) - F(G(n)))^2. \tag{3}$$

If the embedding created for the target and reconstruction is similar, we can assume that the depicted people have similar identities. Other components of the image, like the pose or the background, are largely disregarded[4]. Less relevance is also given to the coloring of the image, including the color of hair and skin. Both depend on lighting conditions - the same person can appear to have a different hair color in a different image. Invariance to color is a desired property in the context of face recognition.

In contrast, for our reconstruction color of hair and skin is an important aspect. Thus, a good starting point for the FaceNet loss training is a $P$ that has previously been trained to minimize the pixel-wise error - the pixel-wise error does not truly motivate similar facial features, but steers the overall image towards the right color composition.

In total, for a target image $X$ and its corresponding output $n$ from P, our loss is given by

$$L(n) = \lambda_p \cdot L_{\mathrm{pixel}}(n) + \lambda_f \cdot L_{\mathrm{facenet}}(n) + \lambda_d \cdot L_{\mathrm{dist}}(n) \tag{4}$$

$$= \lambda_p \cdot (X - G(n))^2 + \lambda_f \cdot (F(X) - F(G(n)))^2 + \lambda_d \cdot \left(\frac{1}{|n|} \sum_{n_i \in n} n_i + |1 - \sigma(n)|\right). \tag{5}$$

The distribution term is always weighted with a fixed $\lambda_d = 0.01$. Our training starts with pure pixel-wise loss ($\lambda_p = 1, \lambda_f = 0$) until the images no longer improve - we denote the resulting model $P$ as $L_{\mathrm{pixel}}$-model. Then the training is switched to facenet loss ($\lambda_p = 0, \lambda_f = 1$), continuing from the $L_{\mathrm{pixel}}$-model, to compute the $L_{\mathrm{facenet}}$-model. Our experiments have indicated that training the $L_{\mathrm{facenet}}$-model with any combination of the two losses slows down convergence and leads to worse optima.

---

[4]We found that even though FaceNet (Schroff et al., 2015) is trained to ignore everything but the identity of the depicted person, images can end up recreating some parts of the pose or background even when trained with nothing but FaceNet embedding loss. This is in line with our main premise: additional information can leak into even very specialized networks.



Figure 3: Stylemixing with a StyleGAN2, between an image $A$ (leftmost image) and an image $B$ (rightmost image). For each of the seven images between $A$ and $B$, all generator inputs are the same as for image $A$, with the exception of a single vector that was taken from image $B$. For the i-th of those seven images, we replaced the i-th input vector with the $B$ vector.

### 3.4 USING THE LAYERS OF A STYLEGAN

A StyleGAN differs from other GANs by its *progressive* training method. Each network block is trained progressively for a different image resolution - starting at rough outlines in $4 \times 4$ images and ending with fine details created by the final block for the highest resolution. The strategy is meant to alleviate the drawback of convolutions and allow a convolutional neural network to model image-wide relations. By starting from tiny images, small enough to be covered by one convolutional kernel, global relations are (said to be) embedded into the lowest layers of the generator.

The training strategy is the reason for a very interesting property of a StyleGAN: each block defines different features of an image, features that can even be distinguished from a human perspective. In our examples, we are using a StyleGAN trained to create $256 \times 256$ images, whose architecture can be separated into seven blocks. Visible in the example in Figure 3, its first block defines the pose of the depicted person, the second and third different facial features, and the fourth the background. Deeper layers are creating smaller details of the image.

This allows to use a $P$ trained for $L_\text{facenet}$ in combination with a $P$ trained for only $L_\text{pixel}$. We feed the output of $P$ trained for $L_\text{facenet}$ to the second and third block, and the output of $P$ trained for only $L_\text{pixel}$ to the rest[5], deliberately assigning $P$s optimized for different loss functions to different aspects of the image.

## 4 EXPERIMENTS AND RESULTS

As the target for our attack, we train a ResNet18 (He et al., 2016) for 300 epochs, minimizing the mean squared distance between the model output and a target attribute vector. The output vectors are made up of 40 or less values, describing abstract properties of both face and image (e.g. "blurriness", see Appendix B.4 for a list of all properties). There are only few publications with the same scenario as our attack, making the most direct competitor the method implemented by Yang et al. (2019). The result of their strategy for our setting is presented in the last row in Figure 5 and 4.

To compare our strategies quantitatively, we are assessing the squared distance (MSE) between images and feature vectors, in addition to the structural similarity index measure (SSIM) (Wang et al., 2004) between images and the similarity between face embedding extracted by VGG-Face (Parkhi et al., 2015) and OpenFace (Baltrušaitis et al., 2016), face embedding networks similar to FaceNet that were not a part of our reconstruction training. Inspired by Tinsley et al. (2021), we also calculate the distance scores between two distinct subsets of CelebA as a baseline, to find that if the images are from the same distribution, but otherwise completely different, the differences are at MSE 0.62, SSIM 0.06, VGG-Face 0.0005 and OpenFace 0.0037.

To the best of our knowledge, there is currently no perfect metric for the visual similarity of two portraits. A standard facial embedding disregards details, like the coloring of the image, while a measure created for the overall similarity of images weights all image areas equally. Additionally, while the statistics are useful to compare images from the same distribution, all are likely to be "fooled" by images from unexpected distributions, like the relatively blurry images from Yang et al. (2019). To compare our images to the previous state-of-the-art, we are instead evaluating the results of a designated user study (see Figure 2): 19 participants are asked to determine the most likely original for a reconstruction, given a selection of five similar portraits (see Section B.2 for details).

---

[5]Both $L_\text{facenet}$ and $L_\text{pixel}$ are always used in combination with $0.01 \cdot L_\text{dist}$

### 4.1 D AND E TRAINED ON THE SAME DATASET

In the scenario of an attack on an encoder that was trained on a public dataset, our $E$ is trained to extract a 32-element attribute vector. The vector describes basic image and facial properties, in addition to the angle of the depicted face (see Appendix B.4 for a full list). Both $E$ and $D$ (i.e. the frozen image generator $G$ and the pregenerator $P$) are trained on FFHQ. To evaluate the attack, we are feeding new images from CelebA to $E$ and then applying our trained reconstruction networks to recreate the original image from the feature vector. Figure 4 presents a qualitative evaluation of nine randomly selected images.

Even though the FaceNet loss started from images that were approximately correct in terms of skin and hair color, the reconstruction is sometimes shifted towards a different coloring over time. The difference becomes especially evident from the fact that the FFHQ dataset exhibits some degree of diversity in terms of skin color, which CelebA is lacking entirely - while all nine random CelebA images are taken from subjects with light skin, some of the FaceNet reconstructions are different. The fusing helps to maintain the overall coloring scheme extracted by the pixel loss, combining them with the more detailed facial features extracted by the embedding loss. Looking at the recreation for the last random target image (last column) reveals that there is a small chance that our process fails entirely, maybe caused by an erroneous feature vector, maybe by an unusual combination of attributes.

The results by Yang et al. (2019) are also recreating basic attributes, however, all reconstructed images are blurry and visually close to the same "average face". While their images usually represent attributes like age and pose correctly, additional information like the shape of the eyes, face and mouth, is barely distinguishable in their images. The images created by our predecessor do come with one advantage: because the images are not bound to the distribution of actual human faces, the scores for common image evaluation methods are quite high, indicating that they could be more useful when attempting to fool a facial recognition software (MSE 0.078, SSIM 0.403, VGG-Face 0.00015, OpenFace 0.0047). Seen in Figure 2, both strategies achieve a maximum user recognition rate of 80%. In terms of average recognition rate and number of false answers ("subtract false answers"), our method outperforms the predecessor.

The metrics shown in Table 1 are in line with our qualitative evaluation: the pixel loss minimizes the difference between images pixels (visible in both mean squared distance and SSIM), while the FaceNet loss minimizes the loss between facial embeddings, though sacrificing pixel accuracy. The combination of both results in almost the same pixel distance as pure pixel based loss training, but with a lower embedding distance. Notable is that the stylemixing significantly decreases the distance between feature vectors - the images are most similar from the perspective of the encoder.

Interestingly, even though hair color is not included in the attribute vector, the reconstruction manages to distinguish "brown" and "not brown" hair with surprising reliability.

### 4.2 D AND E TRAINED ON DIFFERENT DATASETS

For the second setting, we evaluate our attack against an encoder that was trained to extract a 40-element attribute vector on CelebA (see Appendix B.4), while our decoder is still trained on FFHQ. Note that the attribute vectors of this experiment include no information about the position or angle of the depicted face. The reconstruction is evaluated on a subset of CelebA that was never seen by $E$. Hence, $D$ is not only trained on different images than $E$ but even on images drawn from a different distribution.

It is clear from both the quantitative evaluation of randomly selected example images (Figure 5) and the calculated statistics (Table 1 and Figure 2) that this task is more difficult. However, while the pixelwise loss is sometimes unable to create a similar image, the FaceNet loss usually helps the model project matching facial features onto the sometimes otherwise mismatched portraits, proving that this loss formulation is more stable to the challenge of diverging datasets.

The attribute vector used for this reconstruction setting does not include information about the pose of the depicted person. Interestingly, for the images taken from the side, the recreated person is photographed from a similar angle. A more detailed evaluation is presented in the Appendix.

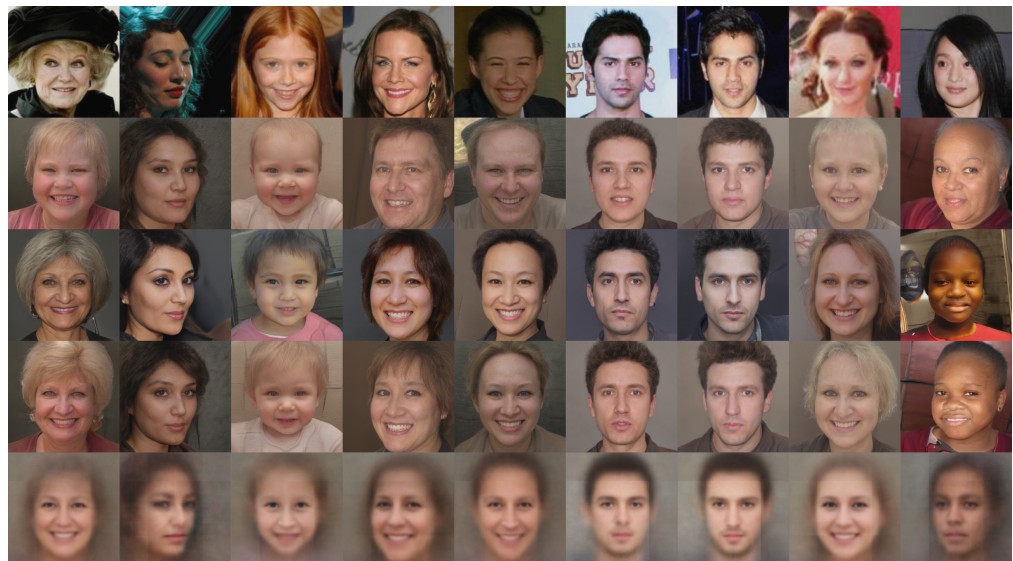

Figure 4: Reconstructions for random images of the CelebA dataset, for a $D$ and $E$ both trained on FFHQ. From top to bottom, the rows present the target images, the result of the pixel-wise loss, the FaceNet embedding loss, and the fused version of the two. The final row displays the outcome of our predecessor, Yang et al. (2019). Details about training can be found in Section 4.1.

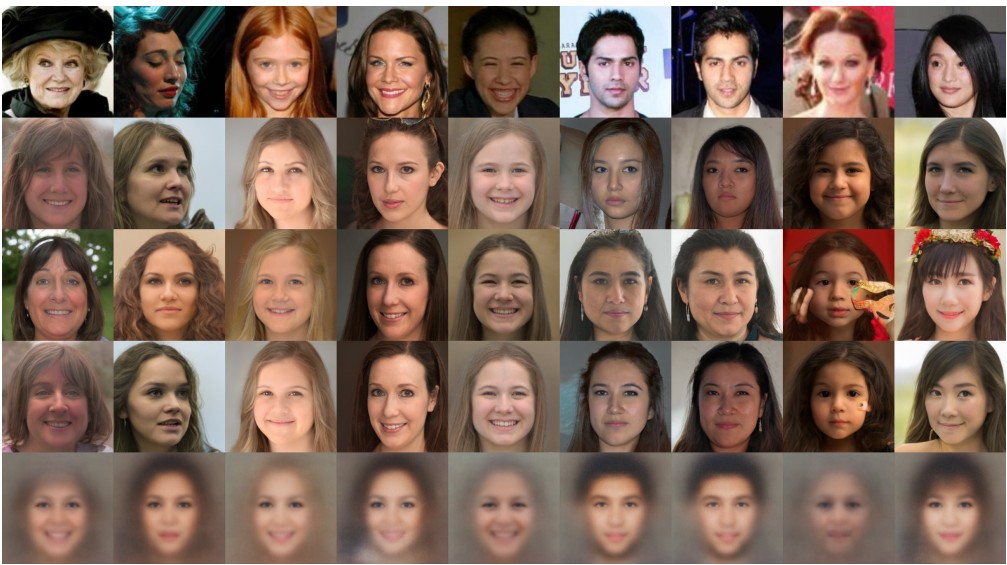

Figure 5: Reconstructions for random images of the CelebA test set, for a $D$ trained on FFHQ and $E$ trained on the CelebA training set. From top to bottom, the rows present the target images, the result of the pixelwise loss, the FaceNet embedding loss, the fused version of the two and the results from Yang et al. (2019). Details about training can be found in Section 4.2.

This information, e.g., is not extracted by Yang et al. (2019): all images are from a frontal perspective, and all are slight variations of an average human portrait, providing much less information about exact facial features than our reconstruction. But, like for the previous setting, scores like MSE and SSIM are higher - possibly because using the "average" for many pixels allows to minimize the loss, despite the absence of distinguishing features (MSE 0.075, SSIM 0.426, VGG-Face 0.00014, OpenFace 0.0050).

Table 1: Quantitative evaluation of our reconstruction of CelebA test images for $D$ trained on FFHQ, and $E$ on FFHQ (top) and the CelebA training set (bottom).

|  |  | MSE | | SSIM | VGG-Face | OpenFace |
|  |  | X | f |  |  |  |
|---|---|---|---|---|---|---|
| FFHQ | $L_{\text{pixel}}$ | 0.3535 | 7.8280 | 0.1200 | 0.0003272 | 0.004061 |
|  | $L_{\text{facenet}}$ | 0.4017 | 4.5761 | 0.1247 | 0.0003367 | 0.003797 |
|  | mixed | 0.3417 | 3.3914 | 0.1302 | 0.0003138 | 0.003723 |
| CelebA | $L_{\text{pixel}}$ | 0.4387 | 0.2059 | 0.0934 | 0.0003947 | 0.003992 |
|  | $L_{\text{facenet}}$ | 0.4399 | 0.1759 | 0.1016 | 0.0003631 | 0.003934 |
|  | mixed | 0.4302 | 0.1744 | 0.0972 | 0.0003967 | 0.004053 |

Table 2: Results of a user study for the two scenarios (see Section 4.1 for the left graph and 4.2 for the right), by our method (blue) and Yang et al. (2019) (orange). The score for an image is determined by the number of correct answers. In the second column, we are counting a correct answer as one, no answer as zero, and a false answer as minus one. See B.2 for a more detailed description of the user study.

|  |  | % correct answers | | % correct - % incorrect | |
|  |  | best image | mean | best image | mean |
|---|---|---|---|---|---|
| FFHQ | ours | 79 | 42 | 79 | 15 |
|  | Yang | 79 | 29 | 73 | -22 |
| CelebA | ours | 68 | 36 | 63 | 11 |
|  | Yang | 53 | 39 | 16 | 5 |

## 5 CONCLUSION

This work introduced a novel reconstruction pipeline, operating with only a low-dimensional vector representing high-level attributes of a portrait and black-box access to the encoder network. The reconstruction surpassed the perceptual quality of previous work in this setting by leveraging recent advances in image generation and facial identification. A pivotal step involved merging the model trained on face embedding loss for mere individual identity with the model from pixel loss for finer image details. This integration was achieved by directing their outputs into distinct layers of the StyleGAN.

Our results show that already a low-dimensional attribute vector and only black-box access to the encoder network can allow unexpected conclusions about the original image - most notable the identity of the depicted person. In a user study we observed recognition rates up to 79%. But also additional details like the angle from which the portrait was taken can be extracted. Despite being intended to encode nothing but a small number of abstract features, the target network inadvertently captures more information from its input than just those features. It's crucial to recognize that a neural network, when employed conventionally, can convey much more information from a high-dimensional input to a low-dimensional output than expected. Consequently, the output of a neural network must be treated with the same level of care as the input.

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

## A  ADDITIONAL RESULTS

### A.1  RECONSTRUCTING THE POSE

Our reconstruction is based on the decoder utilizing both actual correlations in the image data, and correlations that are falsely extracted by the encoder. The exploitation of actual correlations is, for example, apparent in the hair color of the people reconstructed from the FFHQ encoder (Section 4.1): the attribute vector does not provide data for the hair color, but it includes a value defining the overall image brightness. We assume that our decoder bases the hair color on that image brightness attribute. This presumption is correct in many cases, but also means that an image taken in front of a white background usually results in a reconstructed person with blonde hair, independent of their actual hair color.

The existence of false correlations emerges in the successful reconstruction of the facial angle from CelebA attribute vectors, which contain no information purposely describing the position or angle of the face. To assess the pose loss over all test images, we employ a network trained to extract the angle of a face in a comparable manner (Hempel et al., 2022). In their metric, two people looking into the same direction are assigned a pose distance in a range between 0.10 and 0.15, while two people looking into different directions create an output of 1 or more. See Figure 6 for an example of the average values. Since most CelebA images are taken from a frontal perspective, the average pose loss achieved by our reconstruction with a randomly initialized $P$ is already at 0.35. Over the course of our training, that value is lowered to 0.33. The improvement is not significant, it is however too large to be random and proves that the attribute vectors provide at least a hint for the information that should not have been there.

### A.2  RECONSTRUCTING FROM A NOT FULLY TRAINED MODEL

To find a first indication as to whether an information leak is based on an overfitted model, we train our reconstruction pipeline in the same setting as described in Section 4.2, but for an encoder that was only trained for ten epochs. The results seem perceptually similar (see Figure 7), and the image statistics are even slightly better than those of the reconstruction from the fully trained encoder ($f$-MSE 0.12, $X$-MSE 0.39, SSIM 0.11). However, the facial pose is not recreated as well as for the

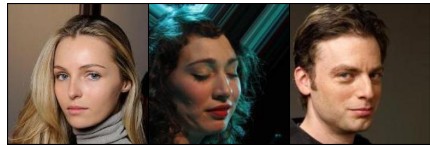

Figure 6: Depiction of the Hempel et al. (2022) pose loss: the pose loss between the left and the center image is 1.1, the loss between the center and right image 0.13.

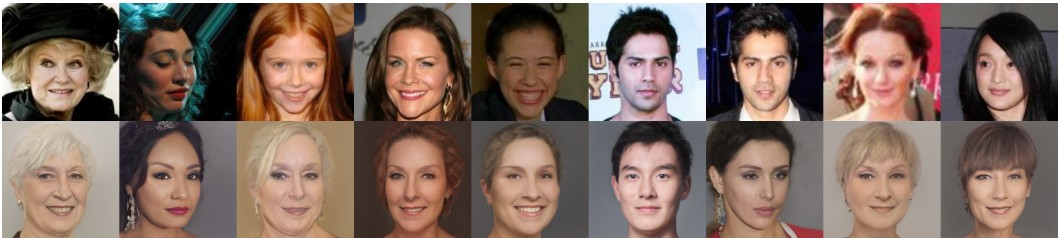

Figure 7: Reconstruction results (fused FaceNet and pixelwise loss output) for $D$ trained on FFHQ, and $E$ trained on CelebA for ten epochs.

preceding experiment. The degree to which the encoder is overfitted to its training set does not seem to affect the overall quality of the reconstruction much, but it might influence the possibility of additional information (like the pose) leaking into the attribute vector.

## B  IMPLEMENTATION DETAILS

The following section summarizes minor details about our implementation which are required to recreate our results, but not relevant to understand the strategy. Our code is available at `https://anonymous.4open.science/r/pge-reconstruction-FCF1/README.md` (for the review, the link points to a temporary anonymous repository).

### B.1  STATISTICS

The quantitative evaluation of our results in Table 1 presents metrics calculated for the entire CelebA testset. Even for the FFHQ setting, in which neither training nor testset contributed to the training, we are only evaluating the test set, to assure the comparability to the CelebA scenario. For the VGG-Face and OpenFace scores, we used the DeepFace library Serengil & Ozpinar (2020). The SSIM calculation is done using *torchmetrics* (Nicki Skafte Detlefsen et al., 2022) (gaussian kernel, $\sigma = 1.5$, kernel size = 11, k1 = 0.01, k2 = 0.03). The exact metric values for the presented qualitative evaluation images are displayed in Figure 8 and 9.

### B.2  USER STUDY

A user study is done in a similar fashion as Fredrikson et al. (2015): we draw a total of 16 random target images from the CelebA test set. For each of them, we then select four more CelebA portraits as *impostors*, depicting different identities with similar attribute values (for the "identifying" items present in both feature vectors, most notably gender, age and hair) - giving us 16 groups of five similar images each. The 16 groups are now randomly assigned to a method (ours or the previous state-of-the-art) and a setting (D and E trained on the same or different datasets), so that we get four groups for each method/setting. A study participant is presented with a reconstruction for the original image, and asked to choose which of the five persons (one original, four impostors) looks most similar, or to select "no answer" if they are unable to identify the most likely original.

Our study had a total of 19 participants. For the default evaluation, each correct answer is counted as one, the rest as zero. For the second method ("subtract false answers"), each correct answer

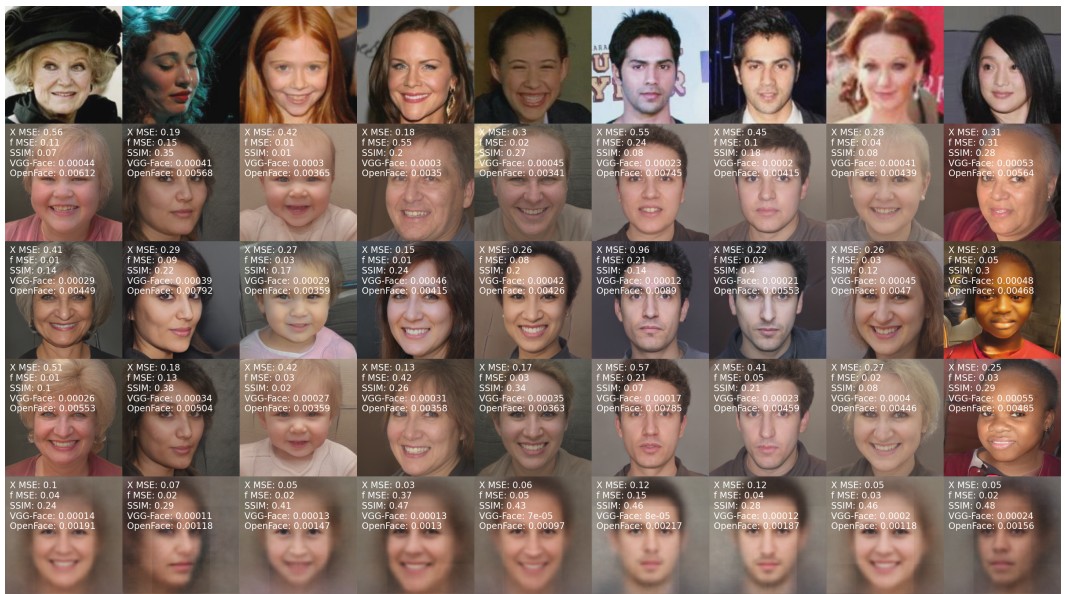

Figure 8: Reconstructions and score values for random images of the CelebA dataset, for a $D$ and $E$ both trained on FFHQ. From top to bottom, the rows present the target images, the result of the pixel-wise loss, the FaceNet embedding loss, and the fused version of the two. The final row displays the outcome of our predecessor, Yang et al. (2019). Details about training can be found in Section 4.1.

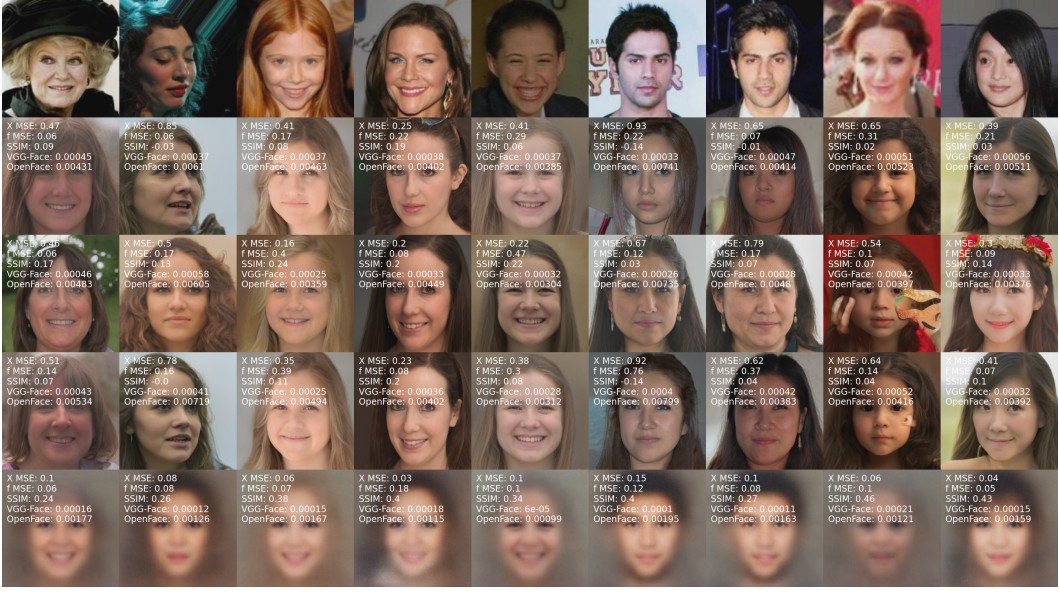

Figure 9: Reconstructions and score values for random images of the CelebA test set, for a $D$ trained on FFHQ and $E$ trained on the CelebA training set. From top to bottom, the rows present the target images, the result of the pixelwise loss, the FaceNet embedding loss, the fused version of the two and the results from Yang et al. (2019). Details about training can be found in Section 4.2.

counts as one, each wrong answer as minus one, and only "no answer" scores as zero. The score is ultimately divided by the number of selections (19 participants $\times$ 4 images per method = 76).

### B.3 P MODELS

Our $P$ is an MLP with the channel dimensions: input channels (32 or 40) $\rightarrow$ 512 $\rightarrow$ 512 $\rightarrow$ 512. Similar to the mapping network of a StyleGAN Karras et al. (2020), each layer (except for the last one) is followed by a Leaky ReLU (slope 0.01). The network is trained with the Adam optimizer (lr=0.0002, betas=(0.9, 0.999), eps=1e-08, weight decay=0), for batch size 64.

### B.4 TARGET MODELS AND ATTRIBUTE VECTORS

Our targets are ResNet18 models, pretrained on ImageNet, with the last layer adjusted to return 32 or 40 items. They have been trained using the Adam optimizer (lr=0.001, betas=(0.9, 0.999), eps=1e-08, weight decay=0) for 300 epochs with batch size 128. The CelebA model achieved 0.0015 and the FFHQ model a 0.2 mean squared distance to the target vectors.

For CelebA, we are using the 40 attributes published by the authors of the dataset. The original target attributes are binary values (1 or 0). We are working from the continous values output of our ResNet18 encoder (which achieved a mean squared distance of 0.0015 to the targets). The provided attributes are:

```
5_o_Clock_Shadow , Arched_Eyebrows , Attractive , Bags_Under_Eyes ,
Big_Lips , Big_Nose , Bushy_Eyebrows , Chubby , Double_Chin , Eyeglasses ,
Goatee , Heavy_Makeup , High_Cheekbones , Male , Mustache , Narrow_Eyes ,
No_Beard , Oval_Face , Pale_Skin , Pointy_Nose , Receding_Hairline ,
Rosy_Cheeks , Sideburns , Young

Bald , Bangs , Black_Hair , Blond_Hair , Brown_Hair , Gray_Hair ,
Straight_Hair , Wavy_Hair

Blurry , Wearing_Earrings , Wearing_Hat , Wearing_Lipstick ,
Wearing_Necklace , Wearing_Necktie

Mouth_Slightly_Open , Smiling
```

As an FFHQ encoder, we are train a ResNet18 to classify a set of 32 features extracted from a public repository[6]. The features that we compress into a target vector are:

```
faceRectangle (top, left, width, height), headPose (pitch, yaw, roll)
, facialHair (moustache, beard, sideburns), emotion (anger, contempt,
 disgust, fear, happiness, neutral, sadness, surprise), makeup (
eyeMakeup, lipmakeup), occlusion (foreheadOccluded, eyeOccluded,
mouthOccluded), smile, age, blur, exposure, noise, hair (invisible,
bald), gender, glasses
```

Note that the haircolor is part of the original attributes, but intentionally left out of this vector.

## C RELATED WORK FOR DIFFERENT SCENARIOS

There are two categories of research that are building on a different set of limitations, but are working towards a similar goal. One is the attempt to reconstruct a representation of a training class, thus compromising the privacy of the dataset used to train a model, the other is a reconstruction from high-dimensional facial embeddings. In the following summation, we are denoting our target encoder as $E$ and the reconstructing decoder as $D$,

### C.1 TRAINING CLASS RECONSTRUCTION

For the recreation of training data, a good representative of one class is assumed to be a private information (e.g. for a facial recognition classifier in which one class depicts one person). The

---

[6]https://github.com/DCGM/ffhq-features-dataset

earliest attempts to reconstruct training classes (Simonyan et al., 2014; Fredrikson et al., 2015) were done for whitebox targets. They are generally starting from a random gaussian noise image $X$, and using gradient descend to optimize that noise with the aim of minimizing:

$$L(E(X), f) \tag{6}$$

where $L$ is a distance measure between feature vectors, usually the euclidian distance. The vector $f$ is the feature vector that is considered to be the "most likely" output for a class - i.e a one-hot encoding of a target class.

Much like recent publications for our chosen scenario, Zhang et al. (2019) and Chen et al. (2020) expand upon the strategy by searching not the actual image $X$, but the input to a GAN trained image generator. Both are training a custom (and comparatively small) image generator in a first step, and then employing gradient descent to iteratively search for the optimal generator input the the second step.

Newer reconstruction methods are oftentimes using only blackbox access to calculate a result. In 2022, Kahla et al. (2022) proposed a method that requires nothing but the class label (not the confidence score) of the target blackbox. Their idea is based on the notion that a "representative" sample for a class is a sample with a large distance to the boundaries of that class.

Published in 2023, Dionysiou et al. (2023) analyse a blackbox scenario, but assume access to the full confidence vector returned by the target model for a posed query. To use the loss from Equation 6 despite the limited access to $E$, they are substituting gradient descent with standard blackbox-optimizers, like CMA. Different from their predecessors, the strategy optimizes over the input space of a variational autoencoder instead of a GAN generator.

Han et al. (2023) build upon the same preliminaries - searching a generator input that minimizes Equation 6 - and implement a reinforcement learning based search strategy to make the search applicable for a blackbox scenario.

### C.2 RECONSTRUCTING FACE EMBEDDINGS

As for the reconstruction of attribute vectors, a fundamental idea to recreate facial embeddings is to minimize the (pixelwise) distance between a recreated and an original image[7]. Razzhigaev et al. (2020) minimize the difference by sampling random Gaussian blobs and adding those that minimize the distance to the recreation. AA common method is also to train a dedicated model on an inverted dataset[8]. Mai et al. (2019) and Shahreza et al. (2022) present a specialized CNN. Mai et al. (2019) expand the loss term by comparing the feature map created by an intermediate layer of a pretrained VGG network - the distance between those feature maps allows to assess the visual similarity of the two images on a different level. The notion is related to our strategy of comparing facial embeddings, though an intermediate layer of a network is much less specific.

Another strategy in which this target formulation oftentimes coincides with others is to search the latent space of a pretrained image generator. Vendrow & Vendrow (2021) and Dong et al. (2023) employ standard blackbox search methods to find the best generator input.

Dong et al. (2021) combine the two ideas, by training a simple MLP to translate a feature vector to a generator output. Their training method is what this paper describes as *noise based training* in Section 3.2, compared to *image based training*, which we consider superior. In a similar approach (Duong et al., 2020), the authors start by training a GAN, inspired by a ProGAN (Karras et al., 2018) (the predecessor of StyleGAN) on a very specific loss function made up of (a) the output of a discriminator and (b) the distance in image-, feature- and latent space to a target image. An interesting component is that they exploit the progressive structure of the generator with the weighting of their loss terms. Early layers of the generator are defining the rough structure of the image and are trained with a larger emphasis on the discriminator output (i.e. how similar the generated images are to a target image set), while the fine details developed in later stages of the training are focused on recreating the identity of specific images. A major drawback of their method is the requirement of training a specialized GAN from scratch.

---

[7]Note that the loss definition is equivalent to the $L_{pixel}$ component in our loss formula (Equation 5).

[8]*Inverted* meaning that the input is a feature map returned by the target $E$ and the expected output (or "label") is the image that created it.

Table 3: Related work in the field of reconstructing data at inference time. VGG16 refers to Simonyan & Zisserman (2014) and face.eoLVe to Cheng et al. (2017). To give a rough idea for the size of CNN target encoders, we name the number of layers, whereas one layer is defined as either a convolution or a fully connected layer, potentially followed by normalization and activation. Note that all training data reconstructions are attacking a identity classifier networks: one class corresponds to one individual, a reconstruction of one of those identities is the targeted privacy breach.

| publication | goal | | setting | | strategy | | target |
|---|---|---|---|---|---|---|---|
| | training data | from model output | whitebox | blackbox | optimization-based | training-based | |
| Simonyan et al. (2014) | x | | x | | x | | CNN (8 layers) |
| Mahendran & Vedaldi (2015) | | x | x | | x | | Dense-SIFT, HOG, CNN (20 layers) |
| Fredrikson et al. (2015) | x | | x | x | x | | Decision Trees, MLP (2 layers) |
| Dosovitskiy & Brox (2016b) | | x | x | | | x | SIFT, HOG, AlexNet |
| Mai et al. (2019) | | x | | x | | x | FaceNet |
| Nash et al. (2018) | | x | | x | | x | CNN (3 layers) |
| Pittaluga et al. (2019a) | | x | | x | | x | SIFT, point clouds |
| Yang et al. (2019) | | x | x | x | | x | CNN (6 layers) |
| Razzhigaev et al. (2020) | | x | | x | x | | ArcFace |
| Duong et al. (2020) | | x | x | x | | x | ArcFace |
| Zhang et al. (2019) | x | | x | | x | | ResNet18 and 152, VGG16, face.eoLVe |
| Zhao et al. (2021) | | x | | x | | x | CNN (4-6 layers) |
| Vendrow & Vendrow (2021) | | x | | x | x | | FaceNet |
| Dong et al. (2021) | | x | | x | | x | ArcFace |
| Wang et al. (2021) | x | | x | | | x | ArcFace (identity classification) |
| Chen et al. (2020) | x | | x | | x | x | ResNet152, VGG16, face.eoLVe |
| Khosravy et al. (2020) | x | | x | | | x | CNN |
| Shahreza et al. (2022) | | x | x | | | x | ArcFace, ElasticFace |
| Kahla et al. (2022) | x | | | x | x | | FaceNet (identity classification) |
| Han et al. (2023) | x | | | x | x | | ResNet152, VGG16, face.eoLVe |
| Dong et al. (2023) | | x | | x | x | | InsightFace |
| Dionysiou et al. (2023) | x | | | x | x | x | CNN (4 layers) |

## C.3 Conditional Face Generation

A different perspective onto the challenge of constructing faces is presented in the context of conditional face generation (e.g. Wei et al. (2022); Yuan et al. (2021)). It is important to separate those methods from our attack. They are aiming to visualize a concrete set of attributes, explicitly steering away from reconstructing additional information from those attribute vectors. We are making use of (possibly unexpected) correlations in the output of a specific encoder, and including the targeted (blackbox) encoder into our pipeline, to reconstruct more information than would be immediately accessible from the attribute vector. For example, we are successfully training to identify hair color or pose, despite the lack of such attributes in the feature vector, while at the same time not reconstructing other attributes like the "blurriness" of the FFHQ feature vector. We are further not introducing any noise into our pipeline, since we do not want to reconstruct "one variation" that satisfies the requirements but, optimally, that one exact image that created the feature vector.

## D Ethical Concerns

As face embedding networks first became popular, their outputs were given no additional protection and were, essentially, treated as an "encoded" version of the input portraits. To this day, a part of the official Microsoft documentation has not been entirely updated: a section about facial embedding

states that "[n]o image of the face is ever stored – it is only the representation"[9]. By now, Microsoft as well as other major companies have recalled that assumption and are protecting and encrypting facial embeddings. This change in strategy is very likely to be attributed to a number of successful reconstructions (for example Dong et al. (2023); Shahreza et al. (2022), see Table 3). Discussing those general privacy issues openly is an important step to making everyone see their relevance.

Our paper is meant to extend that knowledge beyond the scope of facial embeddings. We do not provide means to alleviate the threat and "make feature vectors safe", because in our opinion, there is no way to guarantee that no information leaks into the output of *any* neural network. The output of a neural network requires the same measures of protection as the input.

---

[9]https://learn.microsoft.com/en-us/windows-hardware/design/device-experiences/windows-hello-face-authentication in November 2023

