# OpenReview forum: "Revealing Unintentional Information Leakage in Low-Dimensional Facial Portrait Representations"
_ICLR.cc/2024/Conference — Submitted to ICLR 2024_

### Official Review · Reviewer_5Uwq · 2023-10-30

**Soundness:** 2 fair
**Presentation:** 2 fair
**Contribution:** 2 fair
**Rating:** 5
**Confidence:** 4

**Summary:**

This paper introduces a novel face generation method from a low-dimensional (32 / 40-element) feature vector. The low dimensional feature vector is intended to only describe the attribute information, however the proposed method is able to generate a face that has high similarity with the target image from the encoded feature vector. Thus, it is able to reconstruct the unintentional private information, such as identity. The proposed method utilizes style-GAN and a new loss function based on the face perceptual similarity. In addition, a novel technique to fuse multiple layers of Style-GAN is introduced.

**Strengths:**

+ The proposed method tackles a difficult problem where it wants to reconstruct a target face image from a low-dimensional feature vector. The method works because it effectively designs the framework so that it limits the search space. The feature vector (f) is mapped into a generator input (n) so that the generator (G) is able to produce consistent face image. Thus, the mapping function (P) is trained so that the feature vector can be used to generate similar images as the target image.

+ The loss functions consist of pixel-wise and Facenet embedding losses. While the pixel-wise loss is used to maintain the overall image similarity, the Facenet embedding loss is used to preserve the facial features similarity, such as shape of the eyes, lips, etc. As each block in the StyleGAN defines different features of an image, the mapping function (P) is trained using different losses for each block as mentioned in Section 3.4. Thus, Facenet embedding loss is applied to second and third block which are in charge of the facial features.

+ Experimental results show that the proposed method overcomes the predecessor (Yang et al - 2019) qualitatively. It is able to reconstruct clear and not-blurred face image that has similarity with the target image. In addition, the experiments show that the proposed method achieves low embedding error of the extracted VGG-Face and OpenFace feature embedding. It shows that the generated face has high similarity with the target image.

**Weaknesses:**

- The paper only measures the overall error of the generated dataset. Figure 4 & 5 needs to add the similarity score to give a rough idea about how similar the generated image with the target image. It is unclear how the last column in the figures can be noted as similar person. Note that the paper needs to justify that the private information is leaked in the reconstructed image.

- The paper utilizes different loss functions for each block in StyleGAN. However, there is a lack of justification of the results. The ablation study is required to justify the decision.

- The paper shows that the training of D and E on different dataset is more difficult than on the same dataset. While the quantitative shows the differences, however it is not reflected in qualitative evaluation. In addition, it leads to the question whether the OpenFace and VGG-Face embedding is suitable for measuring the similarity. It is recommended to follow [A] procedure to measure the possibility of identity leakage in the encoder.

- As mentioned in [A], different encoder (face feature generator) might lead to different conclusion in terms of identity leakage. Thus, it is important for the authors to measure the possibility of different face attribute encoder methods. In addition, it is also important to use the more challenging face embedding encoder, like ArcFace or SphereFace to evaluate the similarity between images.

- There are several references that are able to reconstruct face from attribute feature vector [B, C]. Those are might be used for better methods to compare.

Additional references:
* [A] This Face Does Not Exist... But It Might Be Yours! Identity Leakage in Generative Models, WACV 2021
* [B] Attributes Aware Face Generation with Generative Adversarial Networks, ICPR 2020
* [C] Latent Vector Prototypes Guided Conditional Face Synthesis, ICIP 2022

**Questions:**

* What are the similarity scores of the reconstructed and target images in the figures?
* How do the authors justify that the private information is leaked in the reconstructed face image?
* How are the performance using different face attribute encoders?
* How are the performance using different face embedding, such as Arcface and SphereFace?

---

> ### Author Response · Authors · 2023-11-21
>
> Thank you for your time and review. We have now improved our paper according to your suggestions:
>
> * We have incorporated the respective similarity scores into the qualitative evaluation (Figure 8 and 9). Further, we acknowledge that the recreation illustrated in the last column of Figure 4 serves as an example of instances where our method may encounter challenges, and we have included a brief comment on this in the evaluation. It's important to note that the target images were randomly chosen, not cherry-picked. However, we think that the reconstructability of many (though not all) images still poses a potential privacy risk to the subjects depicted.
>
> * Considering that standard image metrics are not always good enough to evaluate the perceptual similarity of faces, we now additionally present the results of a user study (see Table 2) - proving that the reconstructions can be accurate enough to be identified with a high probability.
>
> * Regarding the evaluation method used in [A], we appreciate the merit of demonstrating information leakage by establishing that "the distance between a real image and a generated image is, on average, smaller than the distance between two real images" as suggested by their approach. To align with this concept, we computed the average distance between images from the test and training sets of our target dataset, CelebA, as a baseline. We have included a dedicated paragraph on baseline dataset statistics into the experiments chapter.
>
> * Publication [B] and [C] are examining the topic of conditional image generation, an issue that can appear similar, but is digressing from our research in some key aspects. Seeing that this area is nontheless interesting, especially when delving deeper into the subject, we appended a section about conditional image generation, and the difference to our paper, to the appendix.
>
> We would also like to answer a few of your questions directly:
>
> * Regarding your inquiry about different facial embeddings, we deliberately exclude facial embeddings as a target for our attack within the scope of our research. This  exclusion is due to the considerably higher dimensionality of facial embeddings, explicitly designed to represent the entire identity of the depicted person. Reconstructing an identity from a facial embedding is notably easier and has been repeatedly and successfully accomplished (refer to our related work in Table 2 for examples).
> If you are suggesting that our facial embedding loss could have been based on a different embedding network than FaceNet, your point is certainly valid. FaceNet serves as a robust initial method, given its relatively small network requirements and the availability of reliable pretrained models. While exploring alternative embedding networks and even different image generators than StyleGAN may be a promising avenue for future research, incorporating a more complex embedding network with significantly more training time is not possible within the rebuttal time.
>
> * We make it more clear now in the results section that it implicitely incorporates an ablation study. We showcase the outcomes of our method for (A) solely Mean Squared Error (MSE), (B) FaceNet loss based on MSE, and (C) the combined version. It's crucial to emphasize that (C) is not the result of additional training but rather a new inference method. The intentional demonstration of the combined version (C) outperforming the results obtained by training for the two losses sequentially (B) serves as evidence that our combination method enhances overall performance.
>
> * Indeed, all our attacks are directed at a trained ResNet. While the two ResNets we employed were trained on different datasets, we haven't specifically evaluated potential differences stemming from distinct encoder structures. Analyzing various architectures is undoubtedly a crucial research question. However, it may divert attention from the primary message we aim to convey in our paper: the existence of a widely used network, that is susceptible to unintended information leaks. Even if this vulnerability might not apply universally to all images and encoder networks, the possibility of information leakage, to the extent of identifying individuals, challenges the assumption that an abstract feature vector can be considered a secure representation of an image. The logical next step would be to develop encoders that are resilient against such attacks.

---

### Official Review · Reviewer_3kp1 · 2023-10-30

**Soundness:** 2 fair
**Presentation:** 2 fair
**Contribution:** 2 fair
**Rating:** 8
**Confidence:** 4

**Summary:**

The paper investigate unintentional leakeage from a feature used for face recognition in terms of soft biometrics. The problem is approached using a StyleGAN to build the invertible networks. The method is evaluated on two scenarios in conjunction  with CelebA database.

**Strengths:**

1. The idea to investigate leakage in face recognition modules is very interesting. Approach and results are also welcomed

2. Additional results  from the appendix help to understand better the problematic.

**Weaknesses:**

1. The paper has no "Conclusion" section. While it formally exists, it repeats the main steps of the paper. It is not clear to me what is the lesson learned. The paper is entitled "REVEALING UNINTENTIONAL INFORMATION LEAKAGE ..." and after reading I do not have a clearer view of what has been leaked and what is not

Minor comments:
 - section 2 "Thread" - "threat" ?

**Questions:**

I believe that the paper really needs some conclusion. Otherwise is not complete and therefore it is not clear why people would want to hear from about it. Now the paper is non-committal: there is a problem, we identify it, we analyze the limits of a solution, we propose some solution, we evaluate some solution bearing in mind the limits and this is it.

I expect that starting from the facts [presented in the paper a conclusion can be provided during the rebuttal and I will increase my score. But until then, I believe that the paper is not ready for publication.

------
Post-rebuttal:
----
I have read other reviewers opinion and author rebuttal. In the revised version/rebuttal it has been answered to the question I have asked. While it is not in line with other reviews. I see this paper contribution as being positive. I appreciate the fact that are pointing so some issues which they measure, in my view appropriately. I do not share the opinion that publicizing such aspect could reveal some backdoor for malicious intents.

I am keeping my suggestion which is the paper is interesting and it will attract auditorium.

**Details Of Ethics Concerns:**

None.

Furthermore, the work addresses a problem related to ethical concerns and makes positive

---

> ### Author Response · Authors · 2023-11-21
>
> Thank you for your valuable advice concerning the conclusion. We made a major revision of the conclusion and hope, that the main points of our study become clearer now. Note that we also performed an additional user study to underline our main message: a reconstruction can be accurate enough to identify the subject, even though the attribute vector was not meant to provide such information.

---

### Official Review · Reviewer_nZtS · 2023-11-01

**Soundness:** 1 poor
**Presentation:** 2 fair
**Contribution:** 3 good
**Rating:** 3
**Confidence:** 3

**Summary:**

This paper studies the unintentional information leakage that can happen in deep encoder networks that extract latent representations with abstract attributes from face images. The paper proposes a method that is capable to reconstruct an input face image from a feature vector representation using only black box access to the image encoder. The method is based on the StyleGAN formulation, which is extended with an additional loss that compares the perceptual similarity of portraits by mapping them into the latent space of a FaceNet embedding. The purpose of this paper is to raise awareness about the relevant security issues of existing deep learning systems for face analysis.

**Strengths:**

+ This paper deals with an interesting and important problem that has attracted limited attention from the computer vision community. It is particularly important for reasons related to security and preservation of privacy.

+ The proposed pipeline is intuitive and sound, building upon the formulation of the StyleGAN model.

**Weaknesses:**

- The technical novelty of the proposed method is relatively limited. It only describes a small extension of the loss function of the StyleGAN model. It is mostly interesting as an application of the GAN-based formulations, but I think that it lacks sufficient contributions for a paper accepted in ICLR. Other venues might be more appropriate for such paper.

- The experimental evaluation is highly inadequate. The only quantitative evaluation is the one presented in Table 1. However, this corresponds to an internal evaluation of the proposed method, without any comparison with other SOTA methods. Closely related methods like (Yang et al., 2019) and (Zhao et al. 2021) should have been included in the quantitative comparisons. In addition, a perceptual user study should have been included in the experiments, in order to quantify the performance of the proposed method and other compared methods, in terms of whether the reconstructed faces are perceived by humans to have the same identity as the original real faces.

- The paper has also inadequacies in terms of discussing and citing prior art. First, Some closely-related works, like (Razzhigaev et al. 2020) are only presented in Table 2 of the Appendix. However, such works should have been presented in the main paper, with discussion about their similarities and differences from the proposed method. Furthermore, the paper has not cited some closely-related works like the following:

Khosravy, M., Nakamura, K., Hirose, Y., Nitta, N. and Babaguchi, N., 2022. Model inversion attack by integration of deep generative models: Privacy-sensitive face generation from a face recognition system. IEEE Transactions on Information Forensics and Security, 17, pp.357-372.

Khosravy, M., Nakamura, K., Nitta, N. and Babaguchi, N., 2020, December. Deep face recognizer privacy attack: Model inversion initialization by a deep generative adversarial data space discriminator. In 2020 Asia-Pacific Signal and Information Processing Association Annual Summit and Conference (APSIPA ASC) (pp. 1400-1405). IEEE.

**Questions:**

Please see my comments regarding the weaknesses of the paper.

In addition, there are issues with the clarity of the presentation in some parts of the paper. For example, the last two paragraphs of Section 1 are repeated twice.

**Details Of Ethics Concerns:**

Even though the purpose of this paper is to raise awareness about security and privacy issues of deep learning for face analysis, it could be misused to become a threat for such security and privacy. There is a complete lack of any relevant discussion. The paper should have included such discussion (either in the main body in the appendix), including potential mitigation measures.

---

> ### Author Response · Authors · 2023-11-21
>
> Thank you for your detailed feedback. We have updated our paper to include the suggested improvements:
>
> * “The experimental evaluation is highly inadequate”:
> We have incorporated the quantitative evaluation of (Yang et al., 2019) into our paper. However, a notable issue with their metric is its inclination towards favoring blurred images, which in fact do not provide distinguishing features. This might be missleading and is the reason why we have chosen to present these results in a dedicated paragraph. Assessing the perceptual differences between the two distinct image distributions proves challenging for standard image comparison metrics. We now add a user study, since this seems indeed to be best comparison or even gold standard for the purpose of our work. We appreciate your suggestion.
>
>
> * “The paper has also inadequacies in terms of discussing and citing prior art.”:
> We have chosen to confine the detailed section on related work in the main paper to research conducted within the same setting as our work. Strategies like those presented by (Razzhigaev et al. 2020) deviate fundamentally since they aim to reconstruct from facial embeddings, which are designed exactly for encoding distinctive facial features and are significantly higher-dimensional compared to our low-dimensional attribute vectors. In this respect our problem is much harder so solve. While we acknowledge the significance of publications addressing slightly different requirements, we believe that providing a more detailed description in the main body might divert attention from the primary message of our paper. Nonetheless, recognizing the interest these publications may hold, we have included a more comprehensive discussion of these less closely related papers in the Appendix for the benefit of inquisitive readers.
>
> * Ethics Concerns:
> The Appendix now additionally includes a Section about ethical concerns.
>
> * Concerning the technical novelty:
> We see the key elements of our contribution in two aspects: firstly, the fusion of ensemble outputs through stylemixing, and secondly, the innovation of mapping images into a more representative latent space during the training process. Our research has not revealed prior instances of these concepts in our specific research domain.

---

### Meta-Review · Area_Chair_2sM3 · 2023-12-04

**Metareview:**

This paper proposes a method that converts a low-dimensional feature vector from a black-box encoder into a portrait image. Specifically, given a feature vector from an encoder, it learns a mapping function that maps the vector into the latent space of a pre-trained StyleGAN model, then convert the latent code to a portrait image. The main novelties include: a) introducing StyleGAN as a strong prior, b) incorporating different loss functions and compare the generation and ground truth images in different prior spaces (e.g., face recognition models) and c) specialize the mapping function for different layers in StyleGAN.
This paper raises an important and interesting question about unintentional identity leaking in current portrait auto-encoders. It also proposes a valid solution to prove this opinion. These strengths have been recognized by all reviewers.
However, reviewer nZtS 5Uwq both raised reasonable issues of the submission including: a) limited technical novelty, b) missing evaluations.

**Justification For Why Not Higher Score:**

After reading the paper, reviews and rebuttal, I agree with reviewer nZtS and 5Uwq that this paper is not ready for publication with the following reasons: a) limited novelty. Though StyleGAN has not been employed in this specific task in prior arts, the application of StyleGAN is straightforward in the submission. The learning of a feature mapping network has also been explored in numerous cases in existing works. b) Reviewer nZtS and 5Uwq have both raised issues about the insufficient evaluation of the proposed method. Though the authors kindly include a user study during rebuttal, the participants as well as samples are relatively limited. Thus it is not sufficient to demonstrate the effectiveness of the proposed method. Thus, I incline to reject the paper.

**Justification For Why Not Lower Score:**

N/A

---

### Decision · Program_Chairs · 2024-01-16

Reject